# Modelling of *Escherichia coli* Batch and Fed-Batch Processes in Semi-Defined Yeast Extract Media

**DOI:** 10.3390/bioengineering12101081

**Published:** 2025-10-04

**Authors:** Fabian Schröder-Kleeberg, Markus Zoellkau, Markus Glaser, Christian Bosch, Markus Brunner, Mariano Nicolas Cruz Bournazou, Peter Neubauer

**Affiliations:** 1Department of Bioprocess Engineering, Institute of Biotechnology, Technische Universität Berlin, Ackerstr. 76, 13355 Berlin, Germany; fabian.schroeder@tu-berlin.de (F.S.-K.); mariano.n.cruzbournazou@tu-berlin.de (M.N.C.B.); 2Wacker Biotech GmbH, 07745 Jena, Germany; markus.zoellkau@wacker.com (M.Z.); markus.glaser@wacker.com (M.G.); 3Wacker Chemie AG, 81379 München, Germany; christian.bosch@wacker.com (C.B.); markus.brunner.fe-c@wacker.com (M.B.)

**Keywords:** model-based bioprocess development, *Escherichia coli*, fed-batch, yeast extract, mechanistic model, macro-kinetic model, semi-defined media, complex additives

## Abstract

Model-based approaches provide increasingly advanced opportunities for optimizing and accelerating bioprocess development. However, to accurately capture the complexity of biotechnological processes, continuous refinement of suitable models remains essential. A crucial gap in this field has been the lack of suitable model for describing *Escherichia coli* growth in cultivation media containing yeast extract, while accounting for key bioprocess parameters such as biomass, substrate, acetate, and oxygen. To address this, a published mechanistic macro-kinetic model for *E. coli* was extended with a set of mathematical equations that describe key aspects of the uptake of yeast extract. The underlying macro-kinetic approach is based on the utilization of amino acids in *E. coli*, where growth is primarily influenced by two distinct classes of amino acids. Using fed-batch cultivation data from an *E. coli* K-12 strain supplemented with yeast extract, it was demonstrated that the proposed model extensions were essential for accurately representing the bioprocess. This approach was further validated through fitting the model on cultivation data from five different yeast extracts sourced from various manufacturers. Additionally, the model enabled reliable predictions of growth dynamics across a range of yeast extract concentrations up to 20 g L^−1^. Further differentiation of the data into batch and fed-batch revealed that for less complex datasets, such as those obtained from a batch phase, a simplified model can be sufficient. Due to its modular structure, the developed model provides the necessary flexibility to serve as a tool for the development, optimization, and control of *E. coli* cultivations with and without yeast extract.

## 1. Introduction

Yeast extract (YE) and similar complex media additives are nutrient-rich substrates commonly used to support optimal microbial growth. While predominantly applied during early bioprocess development [1], YE is also utilized in industrial production [2,3]. Its effects on microbial growth have been reported for various organisms including *E. coli* [4,5,6], *Lactobacillus helveticus* [7], and *Saccharomyces cerevisiae* [8] typically influencing biomass formation, growth rate, and acetate production.

Despite this, few studies have mechanistically integrated YE effects into predictive models. Tachibana et al. [9] applied machine learning to predict biomass and product yield in YE-supplemented media, but such models require large datasets, often infeasible in bioprocessing [10,11]. Alternative approaches include semi-empirical models [12], which combine logistic and kinetic equations, and structured kinetic models tailored to specific applications like rifamycin B production [13]. However, these models often rely on simplified assumptions or are too specialized for broader application, and notably, none account for oxygen consumption or the interaction between biomass, limiting substrate, and acetate.

Mechanistic macro-kinetic models [14] provide a more comprehensive framework for representing growth dynamics and have been successfully applied to processes involving *E. coli* [14,15,16,17] and *S. cerevisiae* [18]. These models are also compatible with modern model-based process optimization [19,20,21,22].

Amino acids (AAs), particularly those derived from YE, are key modulators of *E. coli* physiology [4,6]. Their transport and metabolic roles [23,24] and impact on bioprocess performance [25] (pp. 22–23) are well documented. Building on this knowledge, the present study extends the macro-kinetic model of Anane et al. [14] by incorporating additional differential equations to explicitly capture YE effects. The extended model remains computationally efficient and was validated using experimental data from fed-batch *E. coli* K-12 cultivations in YE-supplemented media. Its predictive capability was confirmed across varying YE concentrations and sources.

## 2. Macro-Kinetic Model Formulation

The structure of the model is based on the macro-kinetic description of substrate partitioning by Anane et al. [14]. It extends the functionality by incorporating the growth behavior of *E. coli* when using yeast extract (YE) and a substrate (S), such as glucose, in the batch medium. The model consists of four additional state variables, namely biomass (*X*), acetate (*A*), and dissolved oxygen tension (
DOT
 and 
${\mathrm{DOT}}_{a}$
). These are described by mass balance equations in units of g L^−1^, formulated as ordinary differential equations. All cellular reaction rates related to the state variables are summarized in Figure 1 and were solved as algebraic equations. The description of the model parameters and constants used can be found in the Appendix A.

It needs to be emphasized that the model does not describe the accurate uptake of complex additives such as yeast extract, but rather provides a simplified consumption model derived from the uptake patterns of essential amino acids. The literature shows that the predominant effects of complex amino acid-containing substrates on growth can be represented by three fractions: a rapidly consumed fraction (e.g., serine and aspartic acid), a slowly consumed fraction (e.g., methionine and glutamate), and a nearly unconsumed fraction (e.g., arginine and histidine) [4,23,25].

In this work, the same knowledge was applied to the utilization of yeast extract, which was also described by two fractions: (A) a rapidly consumed fraction (
${\mathrm{YEF}}_{A}$
) with strong influence on metabolism, and (B) a slowly consumed fraction (
${\mathrm{YEF}}_{B}$
) with a presumably weaker influence on metabolism. To account for the rapid and slow utilization characteristics of the yeast extract fractions, it was necessary to restrict the respective parameter bounds, as shown in Appendix A. The initial values of both fractions are determined at *t* = 0 by the distribution
parameter 
$d_{\mathrm{YE},\mathrm{AB}}$
, bounded between 0 and 1 (see Equations (Equation 1) and (Equation 2)). Additionally, a nearly unconsumed fraction (
${\mathrm{YEF}}_{C}$
) can also be accounted for using the distribution parameter 
$d_{\mathrm{YE},\mathrm{BC}}$
: 
(1)
$${\mathrm{YEF}}_{A,0}={YE}_{0}\;\cdot \;d_{\mathrm{YE},\mathrm{AB}},$$

(2)
$${\mathrm{YEF}}_{B,0}=\left({YE}_{0}-{\mathrm{YEF}}_{A,0}\right)\;\cdot \;d_{\mathrm{YE},\mathrm{BC}}.$$


The uptake of both components is modeled by Monod-type kinetics with individual specific uptake rates (
$q_{\mathrm{YEFA}},q_{\mathrm{YEFB}}$
) and affinity constants (
$K_{\mathrm{YEFA}},K_{\mathrm{YEFB}}$
): 
(3)
$$q_{\mathrm{YEFA}}=q_{\mathrm{YEFA},\max}\;\frac{{\mathrm{YEF}}_{A}}{{\mathrm{YEF}}_{A}+K_{{\mathrm{YEF}}_{A}}},$$

(4)
$$q_{\mathrm{YEFB}}=q_{{\mathrm{YEF}}_{B},\max}\;\frac{{\mathrm{YEF}}_{B}}{{\mathrm{YEF}}_{B}+K_{{\mathrm{YEF}}_{B}}}.$$


The time evolution of both fractions and yeast extract is obtained as follows: 
(5)
$$\frac{d{\mathrm{YEF}}_{A}}{dt}=-\frac{F}{V}\;{\mathrm{YEF}}_{A}-q_{{\mathrm{YEF}}_{A}}\;X,$$

(6)
$$\frac{d{\mathrm{YEF}}_{B}}{dt}=-\frac{F}{V}\;{\mathrm{YEF}}_{B}-q_{{\mathrm{YEF}}_{B}}\;X,$$

(7)
$$\frac{dYE}{dt}=\frac{d{\mathrm{YEF}}_{A}}{dt}+\frac{d{\mathrm{YEF}}_{B}}{dt}.$$

where 
${\mathrm{YEF}}_{A}$
, 
${\mathrm{YEF}}_{B}$
, and 
YE
 represent the concentrations of the yeast extract fractions and the total yeast extract in g L^−1^, *V* is the reactor volume (L), and *F* is the feeding rate (L h^−1^). The difference between the total yeast extract and the sum of the fractions after consumption provides insight into the evolution of yeast extract over time (Equation (Equation 7)).

The substrate S (glucose) in the fed-batch process is given as
(8)
$$\frac{dS}{dt}=\frac{F}{V}\;\left(S_{i}-S\right)-q_{S}\;X.$$


The specific substrate uptake (
$q_{S}$
) is described by Monod-type kinetics: 
(9)
$$q_{S}=q_{S,\max}\;\frac{S}{S+K_{S}}.$$


*E. coli* shows increased metabolic activity when yeast extract is added [5], as the amino acids from the yeast extract can be utilized directly instead of being synthesized de novo [24]. As a result, less pyruvate derived from a substrate (e.g., glucose) is used. The phenomenon is represented in this model as non-competitive inhibition of 
$q_{S}$
 (Equation (Equation 10)), where each yeast extract fraction has its individual inhibition strength (
$K_{i}$
).

Further, the concept of acetate cycling [14] was also considered. In contrast to the original model, the oxidative partitioning of the substrate (
$q_{S,ox}$
) is determined first (Equation (Equation 10)), and the remaining glucose flows into the overflow metabolism (
$q_{S,of}$
, Equation (Equation 12)). The maximum uptake capacity of the tricarboxylic acid cycle (TCA) is determined by an inhibition function (
α
, Equation (Equation 11)), where the affinity for the oxidative pathway (
$K_{qSox}$
) depends on the amount of substrate taken up (
$q_{S}$
). The sensitivity of the distribution is determined by 
$K_{qSox}$
, which in theory can assume values between 0 and 100 g L^−1^. The most sensitive region of this parameter is between 0 and 4 g L^−1^, corresponding to a very strong overflow metabolism. Inversely, larger 
$K_{qSox}$
 values exponentially lower the amount of glucose transferred to overflow metabolism. An illustration of this behavior is provided in the Appendix B. This restructuring eliminated the necessity for acetate measurements and simplified the implementation of more complex uptake behaviors, such as those observed in yeast extract: 
(10)
$$q_{S,\mathrm{ox}}=\frac{q_{S}}{1+\frac{q_{{\mathrm{YEF}}_{A}}}{K_{i,{\mathrm{YEF}}_{A},qSox}}+\frac{q_{{\mathrm{YEF}}_{B}}}{K_{i,{\mathrm{YEF}}_{B},qSox}}}\;\alpha ,$$

(11)
$$\alpha =\frac{K_{qSox}}{K_{qSox}+q_{S}},$$

(12)
$$q_{S,\mathrm{of}}=q_{S}-q_{S,\mathrm{ox}}.$$


The formation of acetate, 
$q_{Ap}$
, is a consequence of overflow metabolism: 
(13)
$$q_{Ap}=q_{S,\mathrm{of}}\;\cdot \;Y_{A/S}.$$


The equilibrium (
$q_{A}=0$
) is reached when the acetate produced through the overflow route (
$q_{Ap}$
) is equal to the acetate consumed (
$q_{Ac}$
): 
(14)
$$q_{A}=q_{Ap}-q_{Ac}.$$


The specific acetate consumption rate is modeled as: 
(15)
$$q_{Ac}=\frac{q_{Ac,\max}}{1+\frac{S}{K_{i,A,S}}}\;\frac{A}{A+K_{A}},$$

where it is limited by the acetate affinity constant (
$K_{A}$
) and inhibited by glucose (
$K_{i,A,S}$
) [17]. As in the original model of Anane et al. [14], an external addition of acetate A was not considered in the mass balance: 
(16)
$$\frac{dA}{dt}=-\frac{F}{V}\;A+q_{A}\;X$$


Finally, the remaining substrate, yeast extract, and acetate are used for anabolism (
an
) in the form of a specific growth rate 
μ
 for biomass formation (Equation (Equation 17)). The three components that influence the specific growth rate can be denoted as 
$\mu_{S},\mu_{YE},\mu_{A}$
. The yield coefficient of biomass per substrate (
$Y_{X/S}$
, in g g^−1^) in 
$\mu_{S}$
 is determined excluding cell maintenance (
$q_{m}$
) and is thus given the more correct term 
$Y_{X/S,em}$
. For acetate and yeast extract, individual biomass yield coefficients 
$Y_{X/A},Y_{X/{\mathrm{YEF}}_{A}},Y_{X/{\mathrm{YEF}}_{B}}$
 are determined directly from the respective specific uptake rates 
$q_{Ac},q_{{\mathrm{YEF}}_{A}},q_{{\mathrm{YEF}}_{B}}$
: 
(17)
$$\mu =\underset{\mu_{S}}{\underset{︸}{\left(\left(q_{S,\mathrm{ox}}-q_{m}\right)\;Y_{X/S,em}\right)}}+\underset{\mu_{YE}}{\underset{︸}{\left(q_{{\mathrm{YEF}}_{A}}\;Y_{X/YEF_{A}}+q_{{\mathrm{YEF}}_{B}}\;Y_{X/YEF_{B}}\right)}}+\underset{\mu_{A}}{\underset{︸}{\left(q_{Ac}\;Y_{X/A}\right)}}$$

(18)
$$\frac{dX}{dt}=-\frac{F}{V}\;X+\mu X$$


The energy required for the anabolic metabolic pathways (
μ
) is produced by oxidative energy metabolism (
en
), resulting in a specific oxygen consumption rate (
$q_{O}$
, Equation (Equation 19)). As explained by Xu et al. [17], the mass balance of carbon [
$\mathrm{mol}\_C\;{\left(g\_\mathrm{cells}\right)}^{-1}\;h^{-1}$
] is determined and incorporated to quantify the converted carbon in anabolic and oxidative energy metabolism. Individual oxygen yield coefficients of substrate (
$Y_{O/S}$
), acetate (
$Y_{O/A}$
), and yeast extract (
$Y_{O/YE}$
) are also considered: 
(19)
$$\begin{array}{cc} q_{O}=\; & Y_{O/S}\;\underset{q_{S,\mathrm{ox},en}}{\underset{︸}{\left(q_{S,\mathrm{ox}}-\underset{q_{S,\mathrm{ox},\mu ,an}}{\underset{︸}{\mu_{S}\;\frac{C_{X}}{C_{S}}}}\right)}}\\ & +Y_{O/A}\;\underset{q_{Ac,en}}{\underset{︸}{\left(q_{Ac}-\underset{q_{Ac,\mu ,an}}{\underset{︸}{\mu_{A}\;\frac{C_{A}}{C_{S}}}}\right)}}\\ & +Y_{O/YE}\;\underset{q_{YE,en}}{\underset{︸}{\left(\left(q_{{\mathrm{YEF}}_{A}}+q_{{\mathrm{YEF}}_{B}}\right)-\underset{q_{YE,\mu ,an}}{\underset{︸}{\mu_{YE}\;\frac{C_{YE}}{C_{S}}}}\right)}} \end{array}$$


In accordance with Duan et al. [26], the dissolved oxygen tension (DOT) was reformulated as an algebraic equation to mitigate stiffness and improve numerical tractability. DOT is calculated in percent of saturation with the saturation value of dissolved oxygen (
${\mathrm{DOT}}^{*}$
), the volumetric mass transfer coefficient (
$k_{L}a$
), the Henry equilibrium constant (*H*), and 
$q_{O}$
: 
(20)
$$\mathrm{DOT}={\mathrm{DOT}}^{*}-\frac{q_{O}\;X\;H}{k_{L}a}$$


In order to obtain the measured DOT (
${\mathrm{DOT}}_{m}$
), it is necessary to consider the response of the probe with a static gain of the sensor (
$K_{P}=\tau^{-1}$
), given as the reciprocal of the probe response time 
τ
: 
(21)
$$\frac{d{\mathrm{DOT}}_{m}}{dt}=K_{P}\;\left(\mathrm{DOT}-{\mathrm{DOT}}_{m}\right)$$


## 3. Results

One of the main challenges in developing the mechanistic model is to find the proper trade-off between model simplicity and a sufficiently accurate description of the bioprocess. Since complex components such as yeast extract consist of a large number of substrates (e.g., amino acids, vitamins, proteins, and fats), their influences need to be lumped into the most important effects on the growth rate of *E. coli*, uptake of the limiting substrate, overflow metabolism, and oxygen uptake.

The model can be categorized into four variants: (i) a basic model describing growth on a limiting substrate; (ii) an extended model incorporating yeast extract as an additional carbon source (single-fraction model); (iii) a two-fraction model in which yeast extract is divided into two distinct components; and (iv) a three-fraction model that additionally accounts for an unconsumed fraction of yeast extract. To validate these model concepts with experimental data, a fed-batch cultivation with a recombinant *E. coli* K-12 strain was performed. An initial yeast extract concentration of 10 g L^−1^ in the batch medium was chosen. No further yeast extract was added during the cultivation, and the fed-batch was controlled solely by the limiting substrate glucose. This experimental approach enabled a differentiation of the growth kinetics of *E. coli* with and without yeast extract and provided insights into metabolic responses once yeast extract was depleted. The bioprocess data and the best-fit predictions of the three model concepts are summarized in Figure 2a. The associated RMSE values, representing the goodness of fit, and parameter estimates are given in Table 1. Both the simulations and the high RMSE of the basic model demonstrated that none of the measurable state variables (*X*, *S*, *A*, and DOT) could be adequately fitted, and the error further increased when the entire bioprocess was considered.

This confirmed the necessity of incorporating yeast extract as a distinct carbon source in the model. When the model variants were applied separately to the batch phase and the full process duration, their limitations and differences became apparent. During the batch phase, all models accurately described the measurable states (Figure 2a), with RMSE values differing by only 1%. However, fitting to the entire process revealed notable performance differences. The One-YEF model exhibited a 40% higher RMSE and a visibly poorer fit compared to the other variants, primarily due to uncertainties in predicting acetate formation and substrate uptake. These dynamics were better captured by the Two-YEF and Three-YEF models. In contrast, biomass and oxygen dynamics were equally well described by all models.

The performance differences can be attributed to the underlying cellular dynamics illustrated in Figure 2b. Partitioning yeast extract into multiple fractions allowed the model to flexibly account for both elevated specific growth rates and additional minor effects of complex media components on the metabolic network. For instance, yeast extract fraction A enabled an initial strong suppression of the oxidative glucose pathway, which led to the necessary increase in acetate formation or re-assimilation (
$q_{A}$
). Simultaneously, fraction B slightly enhanced the specific growth rate, supporting biomass formation while facilitating an increase in glucose overflow metabolism (
$q_{S,\mathrm{of}}$
) from 0.0036 to 0.01 g g^−1^ h^−1^ when comparing the Two-YEF to the One-YEF model.

Moreover, when comparing parameter uncertainties between batch and fed-batch processes, the specific glucose uptake rate (
$q_{s,\max}$
)—an identifiable parameter—is particularly noteworthy, differing by a factor of four and simultaneously shifting from 
0.55
 to 
$0.62\;g\;g^{-1}\;h^{-1}$
. A second major change was observed in the distribution parameter 
$d_{\mathrm{YE},\mathrm{AB}}$
, which decreased from 
0.61
 to 
0.27
. These variations in uncertainties and yeast extract–related parameters indicate that, for a more precise determination of yeast extract parameters, determinable quantities such as 
$q_{s,\max}$
 should be known. This information was provided by the fed-batch data.

Overall, the Two-YEF model demonstrated the highest accuracy in capturing bioprocess dynamics involving complex additives such as yeast extract. Incorporating a third fraction did not further improve model performance (Figure 2a) but instead substantially increased parameter uncertainty (Figure 2b, Table 1). Nevertheless, for batch processes or scenarios primarily concerned with biomass formation, the One-YEF model may still provide a sufficiently accurate representation.

### Extended Model Validation

The results show that the Two-YEF model developed in this study was the most sufficient model to describe the growth of *E. coli* supplemented with yeast extract in batch and fed-batch cultivations. The validity of the model was investigated without the availability of amino acid composition data. Focusing on the objectives of the modelling framework, the key features of the model are its capability to predict (i) the effects of variations in the initial yeast extract concentrations on growth and product formation, and (ii) the effect of different yeast extracts on the model parameters. The results are shown in Figure 3b.

First, it was demonstrated that the model can adequately predict growth despite variations in yeast extract concentration in the batch medium up to 20 g L^−1^ by using one parameter set trained with fed-batch data collected at an initial yeast extract concentration of 7.5 g L^−1^. The assumptions incorporated into the model — that yeast extract predominantly affects biomass formation, growth rate, and acetate production — were supported by the experimental data. All three characteristics increased proportionally with rising yeast extract concentrations. For example, when comparing 5 g L^−1^ to 20 g L^−1^ yeast extract, the batch phase duration decreased from 11 to 6 h, while the acetate concentration at least doubled (Figure 3a). Due to volume sensitivities in small-scale reactors, the impact of yeast extract on biomass formation is not distinctly observable. However, in a purely simulated environment, the model predicts approximately a twofold increase in biomass (Appendix B).

Furthermore, it was demonstrated that the previously observed effects persisted across different yeast extracts (Figure 3b). Notably, each yeast extract exhibited a distinct pattern in biomass formation, acetate accumulation, and growth rate. In particular, YE2 displayed a 23% higher biomass yield, a 20% faster growth rate, and a 30% greater acetate concentration compared to the reference yeast extract.

The Two-YEF model successfully captured the process dynamics for all tested yeast extracts, confirming the validity of the proposed mechanistic modelling approach for yeast extract utilization. Moreover, the effects observed were proportional to yeast extract concentration, and the model was able to accurately predict these relationships.

## 4. Discussion

In this work, a model was presented that can accurately describe both *E. coli* bioprocesses with and without yeast extract in terms of the main measurable state variables: biomass, substrate, acetate, and oxygen. Previous yeast extract-based models have considered up to two of these state variables [9,12,13].

The segmentation of process data into batch phase and entire process runtime demonstrated that the simpler One-YEF model is sufficient for accurately capturing the dynamics of batch cultivation. Furthermore, if the focus is limited to biomass formation and substrate uptake during the batch phase, even the original basic model would provide a proper description. Specifically, the basic model would be sufficient if modeling is limited exclusively to biomass formation and substrate uptake in the batch phase and realistic physiological parameter values such as the yield coefficient 
$Y_{X/S}$
 are not considered, which should not exceed a value of 0.5 g g^−1^ [28].

The Two-YEF and Three-YEF models proved effective in accurately capturing all relevant state variables and their interactions across the entire bioprocess. This outcome is consistent with findings from previous studies [4,23,25] and reinforces the macro-kinetic modeling approach presented here, highlighting the importance of classifying yeast extract into at least two substrate groups: highly and poorly metabolizable components. However, in the transition from batch to fed-batch phase, biomass and substrate concentrations were either over- or underestimated in some cases. The addition of an unconsumed yeast extract fraction (
${\mathrm{YEF}}_{C}$
) did not lead to significant improvements. This suggests that a metabolic switch may be triggered by the feed, or that inhibitory effects of yeast extract components such as serine-induced glucose uptake inhibition should be considered [25].

Although the Three-YEF model reflects a more knowledge-driven approach, the absence of quantitative data on the concentrations of individual yeast extract fractions in the bioreactor leads to elevated parameter uncertainty, as multiple distributions of the fractions can yield similar model fits. The integration of advanced analytical techniques, such as Raman spectroscopy, could enable monitoring of these components [29,30], thereby enhancing the reliability of the Three-YEF model and establishing it as a more robust framework for modeling growth on complex media.

Extended tests using different yeast extracts validated not only the proposed model and its consistency with experimental observations in this and prior studies [4,5,6], but also indicated that the model can cover lot-to-lot variabilities, which are a well-known problem in the manufacturing of yeast extracts [2,31,32]. Additionally, the predictive capability of the model was demonstrated under varying yeast extract conditions. This proves the transferability and general validity of the estimated parameters and thus enables the use of model-based optimization and design of experiment methods for bioprocesses involving yeast extract. Moreover, it enhances the reliability and physiological relevance of identifiable model parameters, thereby supporting its application in the characterization of production strains.

## 5. Conclusions

A growth model for *Escherichia coli* in media supplemented with yeast extract was successfully established. By incorporating the concept of amino acid utilization in *E. coli*, additional states were added to describe the significant effects of yeast extract utilization on industrially relevant variables, including biomass, limiting substrate, acetate, and oxygen, and these were integrated into the existing mechanistic model of Anane et al. [14]. The model reliably predicts growth across varying yeast extract concentrations and compositions without any measurements of yeast extract components. Nevertheless, to increase prediction accuracy, non-yeast extract parameters should be determined in the absence of yeast extract, and frequent measurements of biomass and the limiting substrate should be available. Overall, it is well suited for model-based optimization and bioprocess characterization.

In this work, model validation was conducted exclusively for yeast extract in batch media. However, it is likely that the model can also describe bioprocesses in which yeast extract is supplied through the feed solution. Moreover, this modeling approach may be applicable to other complex media additives, although this remains to be verified experimentally in future studies.

## 6. Materials and Methods

### 6.1. Strain, Media and Cultivation Conditions

The cultivations were performed with a recombinant *E. coli* K12 strain, which contains a pMT2 derivative plasmid with a tetracycline resistance. The cells were stored as cryo cultures at −80 °C with an OD_600_ of 1.3 kindly provided by Wacker Chemie AG. The growth medium was a mineral salt medium (MSM) used for the cultivations and precultures as described by Riesenberg and colleagues [33]. The MSM medium was additionally supplemented with 0.1 mL antifoam, and 0.02 g tetracycline per L.

Depending on the objective of the experiments and the mode of cultivation, the initial concentrations of biomass, glucose, yeast extract, and the manufacturer of the yeast extract varied. An overview can be found in Table 2. Experiments without any information about the feed were carried out as batch cultivations. In all experiments, the pH was kept constant at 6.8 and the cultivation temperature was 30 °C. The experiments were carried out in 1 
L
 and 
0.01

L
 scale. The 1 
L
 bioreactor was controlled by a Biostat DCU control unit (Sartorius AG, Goettingen, Germany). Here, the initial stirring speed was 400 rpm and oxygen could not fall below the threshold of 30% due to a stirring control unit. Furthermore, pH was measured using the ERM BIO HB K8 225 pH probe (Hamilton Bonaduz AG, Bonaduz, Switzerland), and DOT was measured using a VISIFERM RS485-ECS 225 H_2_ probe (Hamilton Bonaduz AG, Bonaduz, Switzerland).

The cultivations at 
0.01

L
 scale were carried out in the bioREACTOR48^®^ (2mag AG, Munich, Germany) of the self-driving and automated KIWI-biolab of the Technical University of Berlin. The exact setup of the system can be found in the publications by Haby et al. [34] and Kemmer et al. [35]. Further details about the bioREACTOR48^®^ can be found in Weuster-Botz et al. [36]. The mini bioreactors (MBR) were gassed with 5 
L
/
min
 of an oxygen–air mixture (30% *v*/*v* oxygen) at a stirring speed of 2800 rpm. The gas mixture was obtained by the inflow rates of 
4.43
 Ln/
min
 air and 
0.57
 Ln/
min
 pure oxygen. The evaporation in our facility with an aeration rate of 5 Ln/
min
 air was determined to be 30 
μ

L
/
h
 at 30 °C.

In 1 
L
 cultivations, a periodically adjusted constant feeding profile (“stepwise feeding”) was used as described by Chang [37] and Spadiut [38]. The stepwise feeding profile was implemented with the objective of ensuring an overfeeding regime at the beginning of the fed-batch. The fed-batch process ended after 100 
g
/
L
 of substrate had been fed. In experiments performed in the KIWI-biolab, an exponential feed profile was applied. The exponential feed rate 
$F_{\exp}$
 [L h^−1^] was calculated from the initial feed rate 
$F_{0}$
, the set specific growth rate 
$\mu_{\mathrm{set}}$
 [h^−1^] with 
$\mu_{\mathrm{set}}=0.2\;h^{-1}$
, and the time *t* [h]: 
(22)
$$F_{\exp}=F_{0}e^{\mu_{\mathrm{set}}t},$$

where the initial feed F_0_ was calculated from µ_set_, the glucose concentration in the feed S_i_ [g L^−1^], the yield coefficient Y_XS_, the starting volume V_0_ and the biomass concentration present at the start of the feeding in each MBR X_0_: 
(23)
$$F_{0}=\frac{\mu_{\mathrm{set}}}{S_{i}\;Y_{X/S}}\;X_{0}V_{0}.$$


The pulses of the bolus feed were performed every 5 min. The volume per bolus was calculated as a function of the specified bolus frequency. The raw data from all experiments shown can be found in Appendix B.

### 6.2. Analytics

In the bioREACTOR48^®^, DOT and pH were measured online every 30–60 s by fluorescence sensors (PreSens Precision Sensing GmbH, Regensburg, Germany). The at-line samples were pipetted into a chilled 96 well microwell plate containing dried NaOH per well derived from 15 µL 2 M NaOH, which is dissolved upon addition of the sample leading to an increase in the pH which inhibits cell activity while still avoiding cell lysis. The optical density at 600 nm (OD_600_) was measured on a Microlab^®^ STAR liquid handling site (LHS) (Hamilton Bonaduz AG, Bonaduz, Switzerland) with an integrated Synergy MX microwell plate reader (BioTek Instruments GmbH, Bad Friedrichshall, Germany). Cells were separated from the sample by centrifugation at 15,000× *g*, 10 min, 4 °C. Glucose and acetate concentrations in the supernatant were determined using the Cedex^®^ Bio HT Analyzer (Roche Holding AG, Basel, Switzerland). In the 1 L scale OD_600_ was measured with a DU 730 photometer (Beckmann Coulter, Brea, California, USA) and glucose was measured by a Biochemistry Analyzer YSI 2900 glucometer (YSI Inc., Yellow Springs, OH, USA) and the concentration of acetate in a cell free supernatant (centrifugation 5 min; 13,000 rpm) by Ion Exchange Chromatography.

### 6.3. Parameter Estimation

Simulations and parameter estimations (PE) were performed with a Python (version 3.9) based modelling framework developed in-house. The derivatives are computed using automated differentiation [39] and parameter estimations were performed with lmfit [40]. The generalized non-linear model can be written as
(24)
$$\begin{array}{c} \dot{x}\left(t\right)=f\left(x(t),z(t),u(t),\theta \right), \end{array}$$

(25)
$$\begin{array}{c} 0=g\left(x(t),z(t),u(t),\theta \right), \end{array}$$

(26)
$$\begin{array}{c} x\left(t_{0}\right)=x_{0}. \end{array}$$
 Here, the differential equations *f* correspond to Equations (Equation 5)–(Equation 8), (Equation 16), (Equation 18) and (Equation 21), whereas the algebraic equation set *g* refers to all other equations of the macro-kinetic model formulation (Section 2). The independent time variable is 
$t\in [t_{0},t_{\mathrm{end}}]\subseteq R$
, with 
$x\left(t\right)\in R^{N_{x}}$
 and 
$z\left(t\right)\in R^{N_{z}}$
 denoting the differential and algebraic state variables, respectively. The time-varying inputs or experimental design variables are denoted by 
$u\left(t\right)\in R^{N_{u}}$
, and 
$\theta \in R^{N_{p}}$
 represents the unknown parameter vector.

The model parameters are obtained by minimizing the cost function [41] through a weighted least squares regression applied to all measurement variables 
l∈{1,…,m}
 and sampling time instances 
$k\in \{1,\ldots ,N_{l}\}$
 using the Nelder–Mead method [42]: 
(27)
$$\hat{\theta }\;≔\;\min_{\theta }\;\sum_{l=1}^{m}\sum_{k=1}^{N_{l}}\frac{w_{l}}{\sigma_{l,k}^{2}\;N_{l}}{\left(h_{l}\left(x_{k},u_{k},\theta ,t_{k}\right)-y_{l,k}^{M}\right)}^{2}.$$

The cost function incorporates a user-defined weighting factor 
$w_{l}$
 to adjust the importance of each measurement. For Biomass, Substrate, Acetate, and DOT, values of 
$w_{l}=10,6,4,$
 and 2 were used, respectively. It further considers 
$N_{l}$
, the total number of sampling time instances, the measurement 
$y_{l,k}^{M}$
, and the simulated output 
$h_{l}\left(x_{k},u_{k},\theta ,t_{k}\right)$
 generated using parameters 
θ
. Both measurements and simulated values were scaled using the sklearn RobustScaler [43].

Furthermore, a root mean square error (RMSE) as an additional attribute for the overall goodness of fit was calculated from the residuals of the weighted cost function
(28)
$$\mathrm{RMSE}=\sqrt{\sum_{j=1}^{p}\theta_{j}^{2}}$$


### 6.4. Parameter Uncertainty Quantification

Monte Carlo (MC) simulation was used to calculate variance, standard deviation, and confidence intervals for assessing parameter uncertainties. A total of 
L=250
 replications of the experimental dataset 
$Y_{L}^{m}$
, with 
L=1,…,250
, were generated. To explore the parameter space, Latin Hypercube Sampling (LHS) was applied to each replication to provide a uniformly distributed set of initial samples. For each dataset 
$Y_{L}^{m}$
, parameter estimation was performed to obtain the corresponding point estimates 
${\hat{\theta }}^{1},\ldots ,{\hat{\theta }}^{L}$
.

Due to strong parameter correlations, an initial parameter estimation (PE) was conducted to determine an optimal parameter set (Appendix B). This was followed by a subset selection based on dynamical sensitivities [27], which reduced the parameter set to an identifiable subset. Outliers in the point estimates were removed using the interquartile range (IQR) method, where values falling outside 1.5 × IQR from the first (Q1) and third (Q3) quartiles of the RMSE distribution were excluded.

Finally, the standard deviation 
$\sigma_{\theta_{k}}$
 of the filtered point estimates 
$\hat{\theta }$
 was calculated, and 95% confidence intervals were derived using 
$\sigma_{\theta_{k}}$
 and the Student’s *t*-distribution. The raw data from the MC simulations can be found in Appendix B.

## Figures and Tables

**Figure 1 bioengineering-12-01081-f001:**
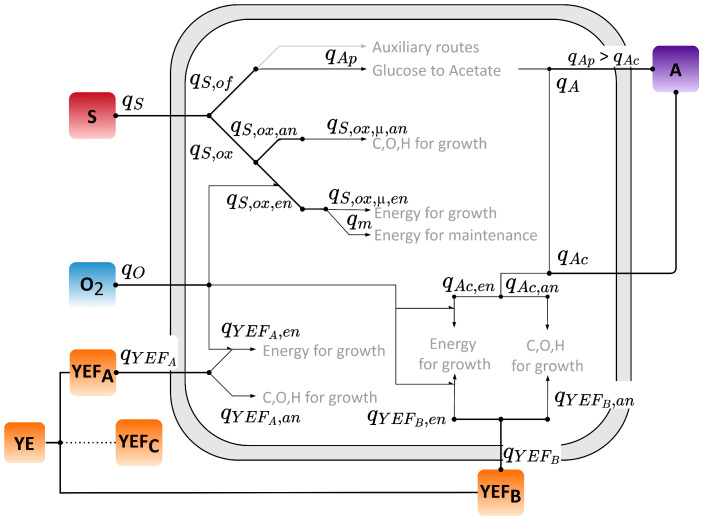
Partitioning scheme of the yeast extract growth model developed in this study. Shown are partitioning of the defined substrate (e.g., glucose), oxygen consumption, acetate formation and re-assimilation and added yeast extract partitioning for modelling substrate (S), oxygen (O_2_), acetate (A) and yeast extract (YE) with the yeast extract fractions A (
${\mathrm{YEF}}_{A}$
), B (
${\mathrm{YEF}}_{B}$
) and C (
${\mathrm{YEF}}_{C}$
) defined in the model. 
${\mathrm{YEF}}_{C}$
, representing the unconsumed portion of YE, is optionally incorporated (dotted line) to enhance consistency with literature-based knowledge. The model differentiates between energetic (en) and anabolic (an) metabolism, and considers both oxidative (ox) and overflow (of) pathways of the defined substrate. To further clarify the macro-kinetic pathways as specific rates (q), each is labeled by purpose: maintenance (m), production (p), consumption (c), and growth (µ).

**Figure 2 bioengineering-12-01081-f002:**
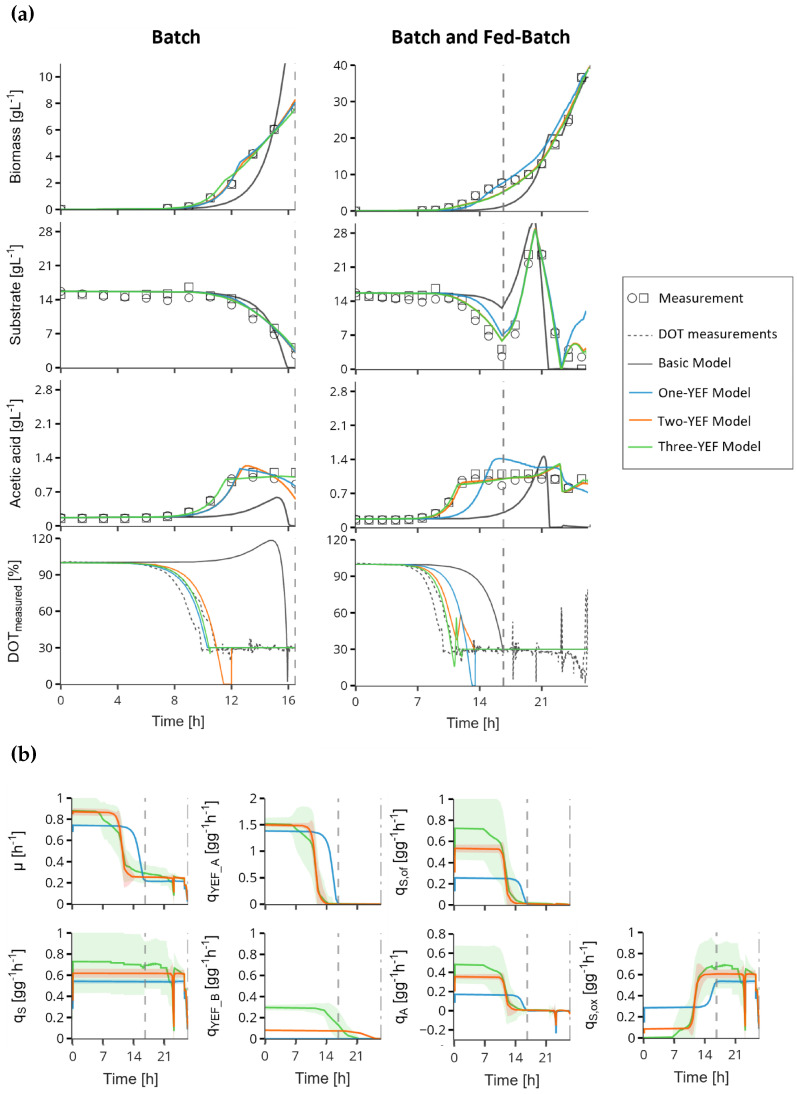
Comparison of model predictions of four different model approaches (lines) and experimental data (markers and dashed lines) from two *E. coli* K-12 fed-batch cultivations conducted in 3 L stirred-tank reactor and supplemented with 10 g L^−1^ yeast extract in batch medium. During cultivation, oxygen was maintained at 30% using a stirring control unit. (**a**) The performance of the basic, the single yeast extract fraction (One-YEF), the two yeast extract fraction (Two-YEF), and the three yeast extract fraction (Three-YEF) model with respect to the state variables biomass, substrate (glucose), acetate, and oxygen in batch phase and over the entire process runtime. Gray vertical dashed lines indicate the feed start. (**b**) The dynamics of the most important simulated specific rates of the modeled metabolic pathways in the One-YEF, Two-YEF, and Three-YEF model over the entire process runtime. Uncertainty bands derived from Monte Carlo simulations are also included.

**Figure 3 bioengineering-12-01081-f003:**
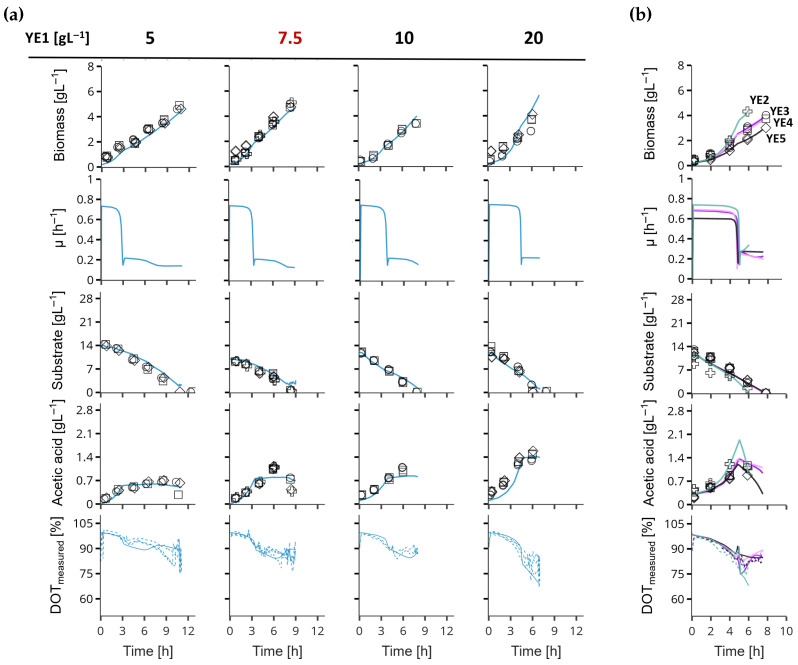
Extended validation of the two-yeast extract fraction (Two-YEF) model. In (**a**), the parameters of the Two-YEF model were trained using data from a batch cultivation supplemented with 7.5 g L^−1^ of the standard yeast extract used in this study YE1 (red) and tested on batch cultivations with 5, 10 and 20 g L^−1^ yeast extract. The corresponding measurement data (markers and dashed lines) and the model predictions with the parameters from the training set are shown. In (**b**), experimental data (markers) of batch cultivations and model predictions (lines) after parameter estimations for each cultivation with yeast extracts YE2 (plus, turquoise), YE3 (circle, purple), YE4 (square, pink) and YE5 (diamond, dark purple) are shown.

**Table 1 bioengineering-12-01081-t001:** Comparison of RMSE and parameter estimates for different model structures. Results of the parameter estimation (PE) with parameter uncertainties for the basic, the single yeast extract fraction (One-YEF), the two yeast extract fraction (Two-YEF), and the three yeast extract fraction (Three-YEF) models. Specific parameters from the literature [17] were kept constant during PE (^1^) or were determined stoichiometrically (^2^). Before quantifying parameter uncertainty, subset selection [27] was applied to identify estimable parameters (^3^). For testing, some non-identifiable parameters (^4^) were also included. The median RMSE from Monte Carlo simulations served as the goodness-of-fit metric.

		Batch				Batch and Fed Batch			
Model		Basic	One-YEF	Two-YEF	Three-YEF	Basic	One-YEF	Two-YEF	Three-YEF
**RMSE**		3.64	2.44	2.40	2.40	1.95	1.10	0.68	0.75
	$q_{S,\max}$ ^(3)^	1.02 ± 0.14	0.46 ± 0.01	0.55 ± 0.19	0.56 ± 0.23	1.62 ± 0.26	0.55 ± 0.05	0.62 ± 0.05	0.71 ± 0.27
	$q_{\mathrm{YEFA},\max}$	—	1.99	1.71	1.48	—	1.80	1.63	1.60
	$q_{\mathrm{YEFB},\max}$	—	—	0.23	0.46	—	—	0.10	0.31
	$q_{m}$ ^(1)^	0.04	0.04	0.04	0.04	0.04	0.04	0.04	0.04
	$q_{Ac,\max}$ ^(4)^	0.70 ± 0.48	0.89 ± 0.47	0.37 ± 0.31	0.84 ± 0.45	0.42 ± 0.47	0.89 ± 0.40	0.55 ± 0.43	0.64 ± 0.45
	$Y_{X/S,em}$ ^(3)^	0.83 ± 0.09	0.33 ± 0.04	0.42 ± 0.07	0.42 ± 0.06	0.42 ± 0.03	0.43 ± 0.02	0.43 ± 0.02	0.43 ± 0.04
	$Y_{X/YEFA}$ ^(3)^	—	0.41 ± 0.01	0.52 ± 0.03	0.50 ± 0.05	—	0.46 ± 0.01	0.56 ± 0.04	0.53 ± 0.11
	$Y_{X/YEFB}$ ^(4)^	—	—	0.14 ± 0.05	0.31 ± 0.14	—	—	0.12 ± 0.05	0.21 ± 0.14
	$Y_{X/A}$ ^(1)^	0.27	0.28	0.30	0.42	0.37	0.50	0.42	0.42
	$Y_{O/S}$ ^(1)^	1.07	1.07	1.07	1.07	1.07	1.07	1.07	1.07
	$Y_{O/A}$ ^(1)^	1.07	1.07	1.07	1.07	1.07	1.07	1.07	1.07
**Parameter**	$Y_{A/S}$ ^(1)^	0.67	0.67	0.67	0.67	0.67	0.67	0.67	0.67
	$Y_{O/YE}$ ^(2)^	—	1.16	1.16	1.16	—	1.16	1.16	1.16
	$K_{S}$ ^(1,3)^	0.05 ± 0.03	0.05	0.05	0.05	0.05 ± 0.03	0.05	0.05	0.05
	$K_{\mathrm{YEFA}}$	—	0.07	0.77	0.05	—	2.99	0.23	0.05
	$K_{\mathrm{YEFB}}$	—	—	1.28	0.10	—	—	1.70	0.10
	$K_{qSox}$	42.16	16.20	65.21	17.69	30.20	80.76	38.11	34.49
	$K_{A}$	0.81	1.00	0.25	0.86	0.66	0.66	0.86	0.86
	$K_{i,\mathrm{YEB},qSox}$	—	—	25.88	49.18	—	—	49.27	49.34
	$K_{i,AS}$	0.28	0.05	2.33	0.08	2.64	0.05	0.05	0.008
	$K_{i,\mathrm{YEA},qSox}$	—	1.13	0.49	0.02	—	1.56	0.24	0.01
	$d_{YE,AB}$ ^(4)^	—	1	0.61 ± 0.11	0.43 ± 0.16	—	1	0.27 ± 0.10	0.21 ± 0.14
	$d_{YE,BC}$ ^(4)^	—	1	1	0.71 ± 0.22	—	1	1	0.54 ± 0.27

**Table 2 bioengineering-12-01081-t002:** Experimental design of all experiments. Abbreviations: YE—yeast extract, OD—optical density, S—substrate, Si—feed stock concentration and init—initial.

Experiments	Scale [L]	OD_600,init_	S_init_ [g L^−1^]	YE_init_ [g L^−1^]	YE Brand	Feed Regime	Feed Strategy	Si [g L^−1^]
Model evaluation	3	0.0002	15	10	Procelys^®^ ^1^	continuous	stepwise	600
YE1	0.01	0.3	9	0, 5, 7.5, 10	Procelys^®^	bolus	exponential	400
YE2	0.01	0.5	12	10	VWR ^2^	—	—	—
YE3	0.01	0.5	12	10	Difco ^3^	—	—	—
YE4	0.01	0.5	12	10	Bacto ^3^	—	—	—
YE5	0.01	0.5	12	10	Roth ^4^	—	—	—

^1^ Lesaffre & Cie, Marcq-en-Barœul, France. ^2^ Avantor Inc., Radnor, USA. ^3^ ThermoFisher Scientic, Darmstadt, Germany. ^4^ Carl Roth GmbH + Co. KG, Karlsruhe, Germany.

## Data Availability

The original contributions presented in the study are publicly available. This data can be found here: https://git.tu-berlin.de/bvt-htbd/public/schroeder-kleeberg_2024_ye_model (accessed on 17 September 2025).

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
