# Peer review of "Modelling of Escherichia coli Batch and Fed-Batch Processes in Semi-Defined Yeast Extract Media"

_bioengineering, 2025, doi:10.3390/bioengineering12101081_

Round 1

Reviewer 1 Report

Comments and Suggestions for Authors

In this manuscript, the authors propose and evaluate mechanistic and unstructured models to describe the kinetics of Escherichia coli (K-12 strain) cultivation in batch and fed-batch modes. The main novelty lies in extending previously published models to account for yeast extract uptake by the cells, either as a whole or through its principal fractions. The extended models demonstrated high accuracy and were validated across different scales (0.01 and 1 L) as well as under varying yeast extract brands and concentrations. Overall, the paper provides a novel and valuable contribution to the field of bioprocess development. Moreover, the manuscript is well-written, clear, and precise. Nevertheless, some major and minor issues must be addressed before the paper can be recommended for publication.

Major comments

i. It is not entirely clear how the authors were able to estimate all parameters related to yeast extract uptake (e.g., total and fractional yields, inhibition coefficients) in the absence of direct YE concentration data, particularly for the Two- and Three-YEF models. I recommend clarifying this point in the manuscript.

Minor comments

ii. Section 2: Please verify that all variables are clearly defined in the text near the corresponding equations.

iii. Page 4, line 93: On what basis is it stated that KqSox is bounded between 0 and 100 g/L?

iv. Page 4, equation 16: If qA is positive (qAp > qAc), acetate should accumulate in the broth medium. Please verify the consistency of the sign for the term qA·X.

v. Page 5, line 111: Shouldn’t ‘Yx/s,em’ be ‘Yx/s,en’?

vi. Page 5, equation 19: Does Cx denote the concentration X? Please standardize the notation used for concentration variables (X, S, A, and YE).

vii. Finally, have the authors investigated the possible composition of fractions A, B, and C of yeast extract? I suggest briefly addressing this point.

Author Response

Dear Reviewer,

Thank you very much for your feedback. Please find all responses to your comments in the attached Word file.

Kind regards,

Fabian Schröder-Kleeberg

Reviewer 2 Report

Comments and Suggestions for Authors

Dear Authors,

I reviewed your highly intriguing manuscript, which presents a new modeling approach for cultivation of E. coli on yeast extract-containing media. The approach is based on an established mechanistic macro-kinetic model for E. coli by Anane et al., which was extended by different mathematical equations in order to address the central aspects of yeast extract uptake. Process variables relevant like biomass formation, concentration of the limiting substrate, acetate, and oxygen, were incorporated. It was shown that thus developed models are of high predictive power regarding biomass formation. Model validation was performed by feeding the model with real cultivation data, based on the use of different lots of yeast extract. It was also shown that the type of model needed to accurately predict the bioprocess strongly depends on the feeding regime – batch processes do with less structured models than fed-batch processes.

The work is scientifically sound, and indeed addresses a crucial knowledge gap, namely cultivation on yeast extract as a complex substrate. The extension of the existing model is comprehensive and understandably described. It is realistic that this approach can be extended to other bioprocesses based on complex nutrient sources.

Only minors should be addressed in order to further improve the manuscript:

Line 83: “de novo” should eb in italics (it´s Latin)

Figure 2a: while the circles are explained (“Measurement”), it is not clear what the squares indicate.

Author Response

(The authors gave the same response as above.)
